# A Multivariate Local Descriptor Registration Method for Surface Topography Evaluation

Chao Kong [1,2], Yuanping Xu [1,*], Zhuowei Li [1], Chaolong Zhang [1], Tukun Li [2], Iain Macleod [1,3], Xiangqian Jiang [2], Dan Tang [1] and Jun Lu [1]

1 School of Software Engineering, Chengdu University of Information Technology, Chengdu 610225, China
2 School of Computing and Engineering, University of Huddersfield, Huddersfield HD1 3DH, UK
3 IMA Ltd., 29 Clay Lane, Hale WA15 8PJ, UK
* Correspondence: ghxpy@hotmail.com

**Abstract:** This paper illustrates a systematical surface topography measurement and evaluation method based on a 3D optical system. Firstly, the point cloud data of the workpiece are extracted by the use of a 3D structured light measurement system, and the STEP file of the design model is converted into point cloud data. Secondly, the local measurement point cloud (LMPC) and digital model point cloud (DMPC) are registered by a multivariate local descriptor registration scheme proposed in this study. Thirdly, the surface shapes extracted from the STEP file are applied as a reference to segment the measuring point cloud. Finally, an error analysis scheme is conducted on specific functional surfaces. An experiment was conducted to analyse the flatness, cylindricity and roughness to demonstrate the effectiveness and advantage of the method. The comparison results show that the proposed method outperforms other 3D optical surface topography analysis methods.

**Keywords:** STEP files; point cloud; surface topography; multivariate local descriptor registration; error evaluation

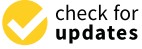



## 1. Introduction

Surface topography is defined as the interface between the part and surrounding medium, which is the concrete reflection of product precision. The form tolerance consists of flatness, cylindricity, roughness, straightness, etc. The assessment of the surface topography is essential for the performance validation of parts. The standardised method for obtaining an evaluation of surface topography is a coordinate measuring machine (CMM). However, the CMMs usually occupy enormous space and are time-consuming. Another approach is to use industrial CT to obtain complete workpiece scanning data and then coordinate measurement data with the CAD data [1,2], which is costly and inefficient.

The use of 3D optical measurement has been widely used in manufacturing, scene modelling and other fields because of its non-contact, fast measurement speed and high accuracy [3,4]. The 3D optical measurement can be divided into the passive method, such as binocular stereo vision, and active methods, such as time-of-flight and digital fringe projection [5]. Compared with other methods, the digital fringe projection has high resolution, accuracy and speed. This study will focus on digital fringe projection-based surface topography error evaluation. Some work has been done for this method via different approaches to improve accuracy. Peng et al. [6] proposed an adaptive grating method to calibrate the distortion error of the projector, which has the advantage of 0.0213 mm (RMS) standard plane measurement accuracy. Yang et al. [7] studied the residual compensation method of projector distortion, this method needs two adjustments to the original raster image, and its standard plane measurement accuracy is 0.0435 mm. Recently, deep learning has been introduced into digital fringe projection. Shi et al. [8] proposed a deep learning method based on enhanced tags and patches for phase recovery. Jeught et al. [9] investigated a neural network with a large number of simulated data sets in the training process,

extracting the height information of objects from a single fringe image. In metrological studies, multiscale methods are used to analyse surface topography. Brown et al. [10] investigated multiscale analyses and characterization to provide correlations between topography and the performance of parts. Peta et al. [11] introduced geometric multiscale methods into electrical discharge machining; it is possible to shape surface microgeometry. Kang et al. [12] presented a contact model to describe normal and tangential contact behaviors of rough interface with a multiscale method; this model provides an effective way for studying the contact response of rough interface.

The 3D structured light system has the advantages of simple structure and low cost, hence quickly obtaining the measurement data of a workpiece [13]. In practice, the result of optical measurement data is in the form of a point cloud. Take a grating structured light measurement system as an example [14]; it processes the shot raster image to obtain the 3D point cloud of a workpiece, known as the measurement model.

The classical error analysis method of 3D optical measurement calculates the distance between point pairs after completing the registration of the measurement model and corresponding CAD model. However, obtaining the entire measurement model requires a splice point cloud from multi-view because a point cloud can only photograph the data at a particular projection direction [15], which is high-cost and has splicing errors. To overcome the difficulty of error analysis from local data to whole CAD data, the common method in the industry is to post positioning markers [16]. Nevertheless, it is evident that this method will increase the time and introduce manual intervention errors. Xie et al. [17,18] investigated multiple global descriptors to enhance the description of the point cloud formed by the key points and their neighbourhood to complete the localisation of aircraft partial skin. However, for some small parts, the data is obtained at a specific projection direction, which accounts for a large proportion of the whole digital model, such extensive local measurement data is hard to be broken into patches of the point cloud. The local descriptor method [19] can complete small parts to complete registration, and the commonly used point cloud-based local descriptors depend on manual processes, such as Point Feature Histogram (PFH) [20], Signature of Histogram of Orientation (Shot) [21] and Ensemble of Shape Functions (ESF) [22]. These descriptors are challenging to accurately describe the characteristics of particular parts and have low generalisation.

In addition to the global deformation analysis of the local point cloud, another critical issue to be addressed in implementing target surface error analysis is to split the point cloud into various functional surfaces accurately. Therefore, the research on the segmentation of measuring point clouds of some workpieces is still in the exploratory stage, one of the reasons is that the functional surface is related to the mutations, but some functional surfaces can be connected naturally and smoothly. According to these properties, Qie et al. [23] employed conformal geometry to segment the mesh model of measured data and further studied logistic regression to smooth boundaries. However, the threshold values are difficult to generalise in this method for different parts. Yang et al. [4] used the initial graphics exchange specification (IGES) file as a template to segment CT scan data for error analysis of interested surfaces; however, IGES files are large, and the description of some geometric entities is unstable [24].

In summary, there are two problems to be solved in the error analysis of 3D optical measurement data:(1) how to accurately complete the local point cloud to the whole CAD model registration; (2) how to isolate the target surfaces on a local measurement model. To this end, a multivariate local descriptor registration method is introduced to solve the registration problem in this study. The aim of this method is to establish a complete part-in-whole point cloud registration process, including coarse and fine registration, and improve registration accuracy. Then, to avoid unnecessary functional surfaces and computational power, a local point cloud segmentation method to isolate target surfaces is presented. This study aims to illustrate the entire process of 3D structured light measurement and demonstrate the advantage of the method. Figure 1 shows the framework of the implementation of the error analysis process. Firstly, the standard for the exchange of product

(STEP) digital model is transformed into a point cloud. Then a multivariate local descriptor is used to complete the registration of the local point cloud to the STEP digital model. After registration, the STEP file is used as the reference to segment each functional surface in measurement data. Finally, the error analysis of the target surfaces is completed.

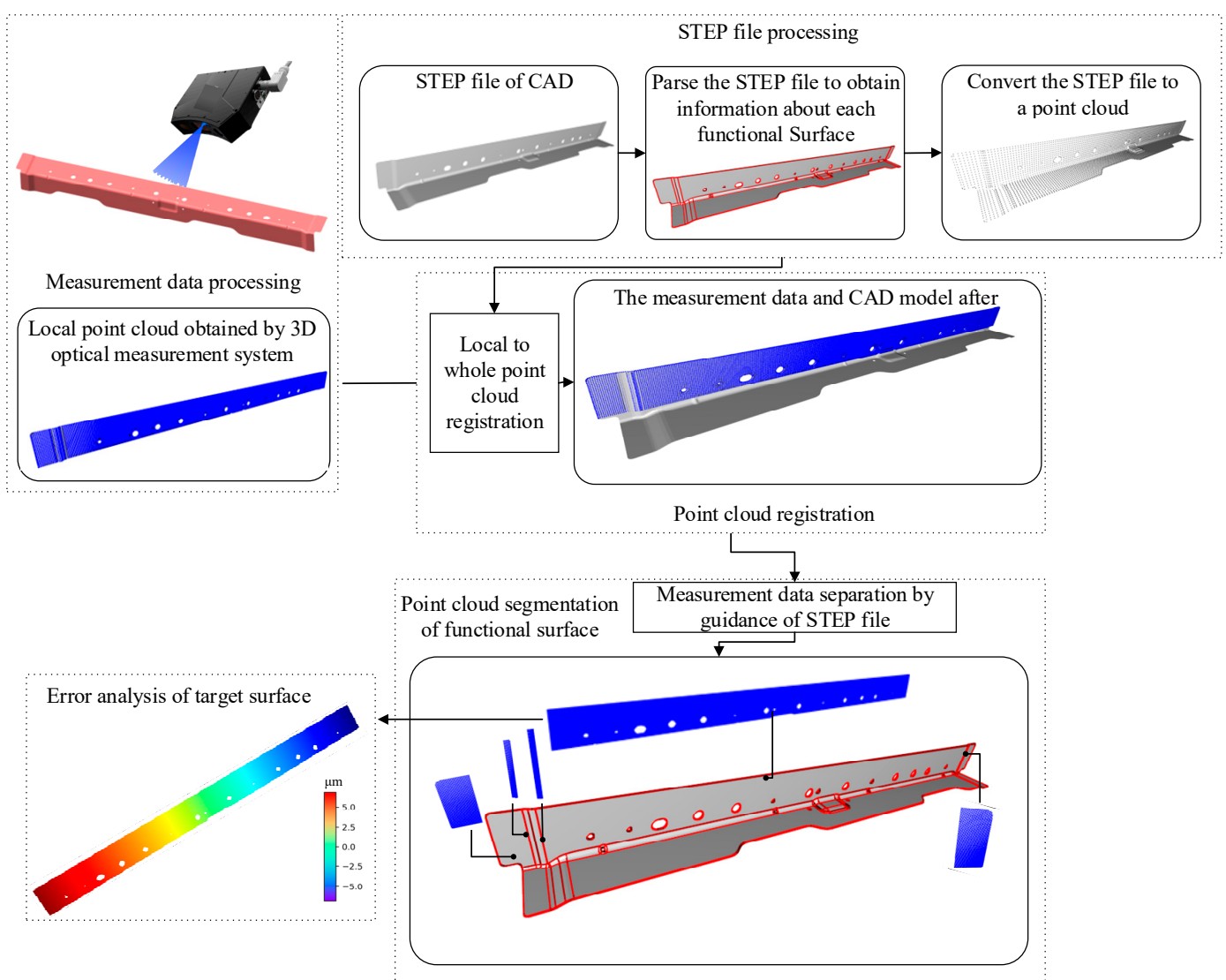

**Figure 1.** The flowchart of surface topography analysis (μm).

This paper is organised as follows: Section 2 illustrates the details of generating functional surfaces and point clouds based on STEP files. The construction of local point cloud registration to the STEP model based on multivariate local descriptors is described in Section 3. Section 4 presents the segmentation method for the local point cloud. Experimental verification of the application and demonstration of its effectiveness and advantages are reported in Section 5. A comparison of different evaluation methods is presented in Section 6. Finally, Section 7 concludes this paper and elaborates on future studies.

## 2. Generation of Functional Surfaces and Point Cloud Based on STEP File

STEP has been developed by ISO committees to describe a complete product definition throughout the life cycle of products [25]. In addition, the STEP file contains useful geometric and topological information [26], and all surface shapes are conformable to the initial design intentions of CAD designers. Therefore, the STEP file can be used as a guide

to segment the measurement data. This section introduces the basic concepts of generating various functional surfaces and point clouds based on the STEP AP242 [27] format file.

### 2.1. The Division Functional Surface in STEP File

The target surfaces in STEP are defined as functional surfaces that contain the demanded geometric features, such as matching surface and *R* angle. An *ADVANCED_FACE* in a STEP file represents a functional surface that includes the type and boundary information based on boundary representation (BRep) [26]. The entity information of a surface is shown in Figure 2. A surface contains multiple edges, and the entity of an edge is divided into lines, circles, rational b-spline curves with knots and non-uniform rational b-spline (NURBS) curves, and the line matrix is defined by

$$p = \begin{bmatrix} 1 & t \end{bmatrix} * \begin{bmatrix} o & d \end{bmatrix}^T \tag{1}$$

where *t* is the length parameter of the line, *o* is the origin point, and *d* is the direction. The circle equation is expressed as

$$p = \begin{bmatrix} 1 & r\cos\theta & r\sin\theta \end{bmatrix} * \begin{bmatrix} o & a & b \end{bmatrix}^T \tag{2}$$

where *r* represents the radius, $\theta$ represents the angle of the circle, *o* represents the centre of the circle, and *a* and *b* are vectors perpendicular to each other and perpendicular to the normal direction of the circle. The NURBS curve equation is given in the form

$$
\begin{aligned}
p(k) &= \frac{\sum\limits_{i=0} N_{i,m}(k)w_i P_i}{\sum\limits_{i=0} N_{i,m}(k)w_i} \\
N_{i,0}(k) &= \begin{cases} 1(k_i \le k \le k_{i+1}) \\ 0(others) \end{cases} \\
N_{i,m}(k) &= \frac{(k-k_i)N_{i,m-1}(k)}{k_{i+m}-k_i} + \frac{(k_{i+m+1}-k)N_{i+1,m-1}(k)}{k_{i+m+1}-k_{i+1}}, m \ge 1
\end{aligned}
\tag{3}
$$

where *N* represents the basis function, *w* represents weight, *P* is the control point, *m* is the order, and *k* indicates the parameter between knots.

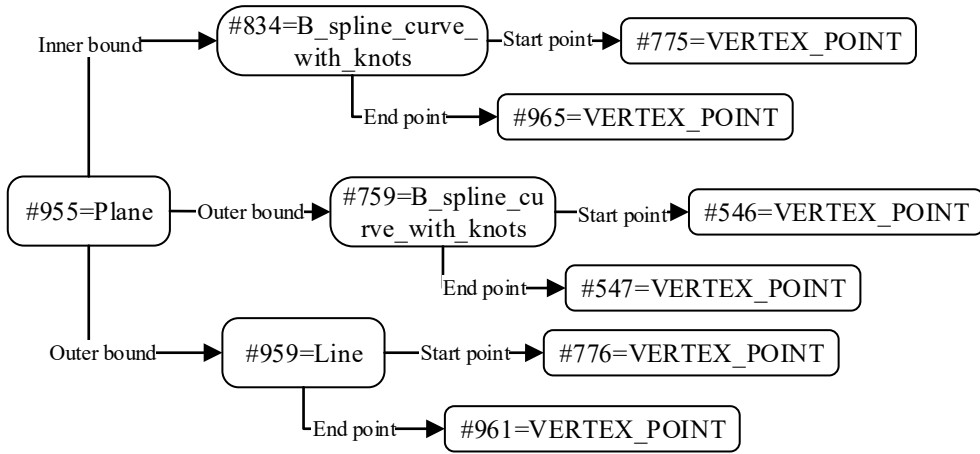

**Figure 2.** The entity information of a particular functional surface in STEP file.

The analysis of circular arc is key in STEP parsing. Although the start and end points are given in the STEP file, it is necessary to convert them into angles through specific methods. Usually, the calculation of the cosine angle of two points can only get the range from 0 to $\pi$ but lacks the range from $\pi$ to $2\pi$. To this end, the content of the circle calculation is given as follows:

(1) Calculating two direction vectors perpendicular to each other in the normal direction *N* of the circle. Selecting any one of the *x*, *y*, and *z* directions of the global coordinates

and taking the cross product with *N*. If the result is not 0, set it as direction *a*; otherwise, choose another direction to calculate the cross-product. Keep taking *N* to cross *a* and direction *b*, obtaining *a*, *b* and *N* as the local coordinate system of the circle, and then projecting it into a 2D plane in the direction of *N*, as shown in Figure 3; the transformation formula is expressed as Equation (4), where $R = [a, b, N]$, *p* is the 3D coordinate point, $\hat{p}$ is the 2D coordinate point, and here is the start point or end point, *o* is the centre of the circular arc.

$$\hat{p} = R_p - R_o \tag{4}$$

(2) In order to remove the *z*-axis of the local coordinate system, the start and end points of the two-dimensional plane can be obtained and calculate the sine $v_1$ and cosine $v_2$ of the starting point *S* and the ending point *E*, respectively.

(3) The equation obtains the results of the final start point and end point (5). Finally, the direction of the edge curve in the STEP file is used to determine the result (if the direction is *T*, select $\theta_1$ in Figure 3, otherwise, select $\theta_2$ in Figure 3). The discrete points of a circle can be determined according to the equation of a circle.

$$\theta = \begin{cases} \arccos(v_1), v_1 > 0, v_2 \geq 0 \\ \arccos(v_1), v_1 \leq 0, v_2 \geq 0 \\ \arccos(v_1) + 2[\pi - \arccos(v_1)], v_1 < 0, v_2 < 0 \\ \arcsin(v_2), v_1 \geq 0, v_2 < 0 \end{cases} \tag{5}$$

According to the parse of boundary, each functional surface is finally divided, as shown in Figure 4.

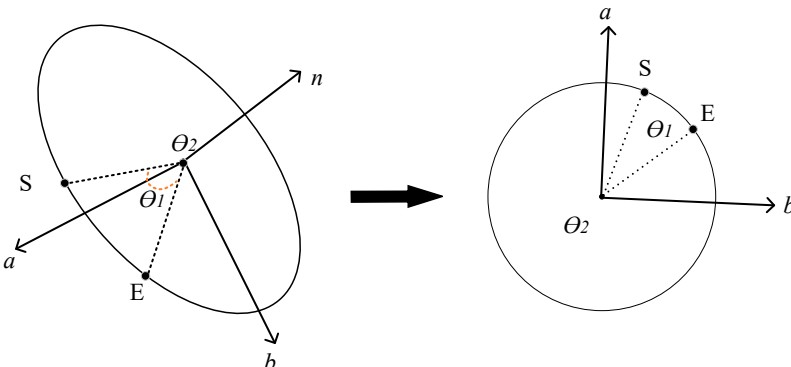

**Figure 3.** The definition of angle in a circular arc.

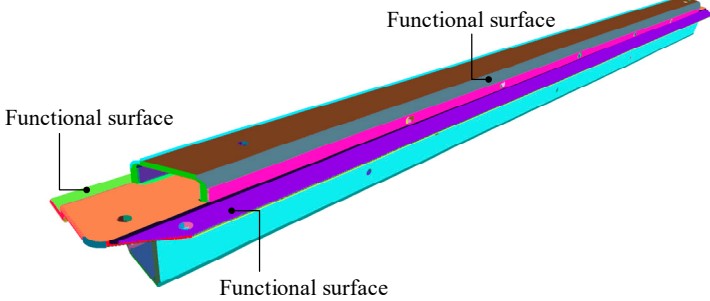

**Figure 4.** The division of functional surface in a certain STEP file.

## 2.2. The Generation and Trimming of Point Cloud

### 2.2.1. The Filling and Trim of Planar Point Cloud

The steps of plane point cloud filling are set as follows:

Input 3D model point cloud;

Output Trimmed point cloud of the plan.

Step 1 Projecting onto a 2D plane and filling the bounding box. The ray casting method is used to determine whether the point is inside the plane.

Step 2 Projecting onto a 2D plane continues with the projection method by Equation (4).

Step 3 Draw the bounding box of the projected polygon and equidistant lines in the bounding box for filling.

Figure 5 shows the result of a polygon filling. After filling, it is necessary to determine whether the filled point is inside the enclosing polygon. Therefore, the ray casting method is adopted for judgment, and the schematic diagram of the ray method is shown in Figure 6. In the ray method, drawing a ray from the point which needs to be determined, if the number of edges that the ray passes through is even, the point is outside the polygon (Figure 6, Point 2). Otherwise, the point is inside (Figure 6, Point 1). When the edge of a polygon is a smooth curve, it can be approximated as the edge formed by multiple straight lines, as shown in Figure 6b.

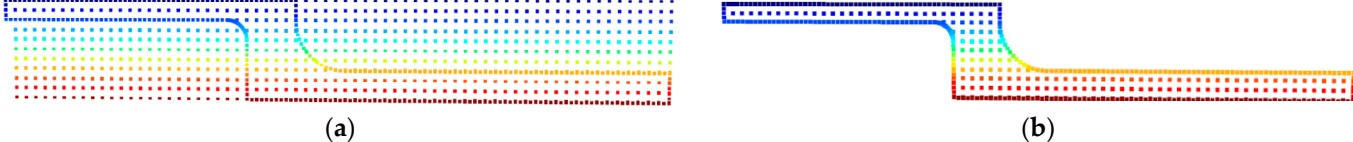

(**a**)　　　　　　　　　　　　　　　　　　　　　　(**b**)

**Figure 5.** The procedure of plane point cloud trimming: (**a**) The point cloud of the plane before trimming; (**b**) The trimmed point cloud through the ray casting method.

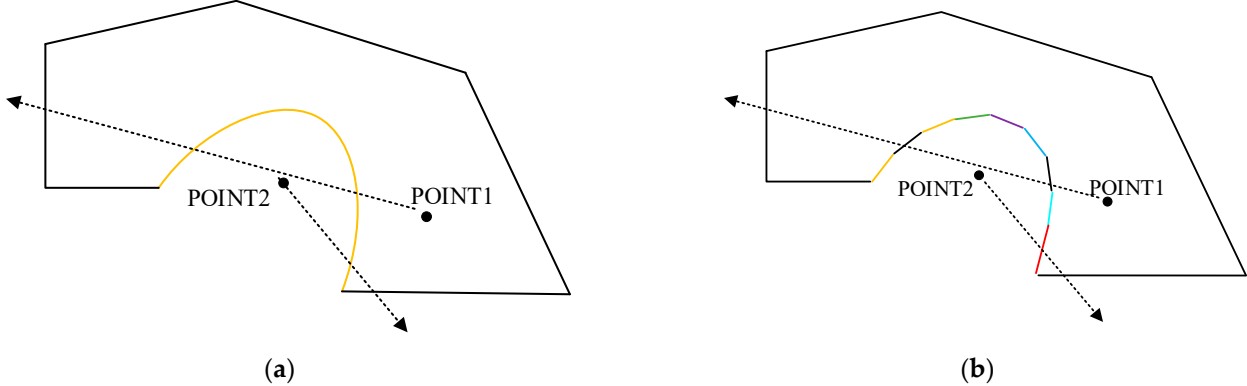

(**a**)　　　　　　　　　　　　　　　　　　　　　　(**b**)

**Figure 6.** The illustration of the ray casing method. (**a**) A polygon includes a curve; (**b**) A curve is reduced to a polygon with multiple straight lines.

2.2.2. The Filling and Trim of the Curved Surface Point Cloud

Compared with a plane, the curved surfaces in STEP files consist of cylindrical surfaces, conical surfaces, b-spline surfaces with knots and non-uniform b-spline. In this study, all curved surfaces were transformed into NURBS surfaces [28]. Similar to the planar point cloud process, the point cloud of curved surfaces needs to be trimmed. It is difficult to trim surfaces in a 3D form directly, and it also cannot follow the trimming steps of the plane, which is trimming by mapping into the 2D space domain. NURBS surfaces can be trimmed by reverse-mapping the boundary to the 2D parameter field of NURBS surfaces. As shown in Equation (6), each point in the 3D domain corresponds to the $u$, $v$ parameter of the NURBS surface. Supposing a point $P(x, y, z)$, a BFGS method can be used to approximate its corresponding parameters $u_0$ and $v_0$ in the parameter domain. The BFGS method [29,30] is a quasi-newton method. Thus, a better initial value of the parameter should be chosen to avoid calculating a wrong result. The trimming steps of the curved surface point cloud based on the BFGS method are as follows:

Input Curved point cloud.

Output Trimmed curved point cloud.

Step 1 Supposing a set of boundary points [31], find their nearest points in the NURBS surface and take their corresponding values $u_i$ and $v_i$ in the parametric domain as the initial values of the Newton iterative method.

Step 2 To define the objective optimisation function:

$$l(u, v) = [p(u, v) - P][p(u, v) - P]^T \tag{6}$$

When $l$ equals zero, the $P$ is obtained as the optimal value in the parametric domain. Then, determine the initial parameter $x_0 = [u_0, v_0]^T$ and initial a symmetric positive definite matric $B_0$, which is the identity matrix.

Step 3 Computation of the first-order partial derivative:

$$g_0 = \begin{bmatrix} l_u(u_0, v_0) \\ l_v(u_0, v_0) \end{bmatrix} \tag{7}$$

Step 4 Calculation of the $x_{i+1}$:

$$x_{i+1} = x_i + \lambda_i B_i^{-1} g_i \tag{8}$$

where $\lambda_i$ denotes the step size, continue to calculate the $s_i = x_{i+1} - x_i$, $y_i = g_{i+1} - g_i$. Computation of the $B_k$:

$$B_{i+1} = B_i + \frac{y y^T}{y^T s} - \frac{B_i s s^T B_i}{s^T B_i s} \tag{9}$$

Step 5 Repeating the above step until the $\|g_{i+1}\| < \varepsilon$.

The optimal parameters $u$, $v$ in $P$, and the value of boundary points in the parameter domain can be obtained according to the above steps; the parameters can be clipped by the ray casting method, and then the clipped parameters can be converted into the spatial domain by Equation (10), and the clipped surface can be obtained. In Equation (10), the $u$ and $v$ represent the parameters in a different direction; the $N$ is the basis function whose details are in Equation (3), $w$ is the weight, $P$ is the control point, and $m$ and $n$ are the order in a different direction. Figure 7 shows the flowchart of the trimmed NURBS surface.

$$p(u, v) = \frac{\sum\limits_{i=0}\sum\limits_{j=0} N_{i,m}(u) N_{j,n}(v) w_{i,j} P_{i,j}}{\sum\limits_{i=0}\sum\limits_{j=0} N_{i,m}(u) N_{j,n}(v) w_{i,j}} \tag{10}$$

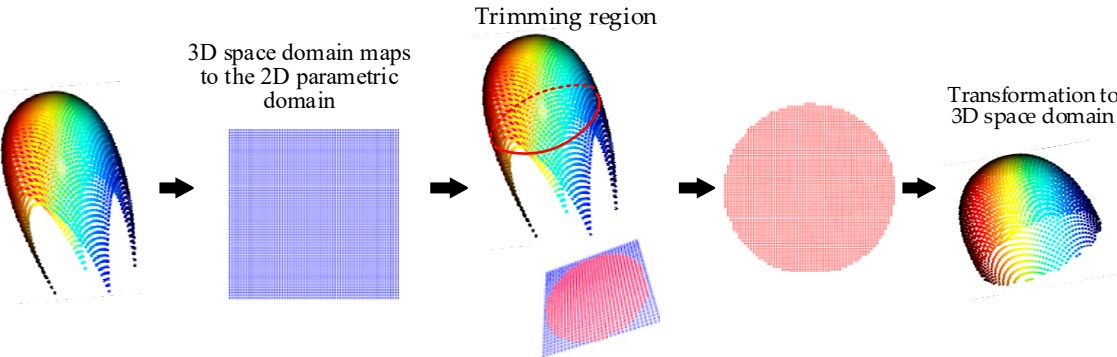

**Figure 7.** The trim of the curved surface point cloud.

## 3. Part-in-Whole Point Cloud Registration

This section is to recognise local geometric features on a complete CAD model and realize the registration of the local point cloud and CAD model, which is the basis of error analysis [17].

### 3.1. The Patch of Point Cloud

In this section, the random sphere cover sets (RSCS) method is adopted to improve the coverage of the whole point cloud [32]. The original RSCS method increases the matching accuracy of subsequent point cloud descriptors. However, the sphere radius associated with this method is fixed each time, which leads to the uncharacteristic patch obtained by a fixed radius. Therefore, the multi-scale RSCS is proposed in this study to increase the diversity of the radius scale of points to improve the matching probability between corresponding point clouds. The multi-scale RSCS division process of the point cloud is as follows:

Input DMPC and LMPC.

Output Multi-scale random sphere cover set and central points of DMPC and LMPC.

Step 1 Down sampling DMPC and LMPC.

Step 2 Randomly selecting a point as the centre of the sphere and dividing the point cloud via radius $r_i$, all point clouds within the radius are identified as the point cloud of the first sphere $s_1^i$.

Step 3 Randomly select a point that does not belong to a sphere $s_k^i$ as the next centre point and obtain the next sphere with $r_i$.

Step 4 Repeat step 3 until every point is covered sphere.

Step 5 Resetting the radius and repeat step 4.

According to the above steps, the multi-scale random sphere cover set of the DMPC $p = \{s_1^i, s_2^i, \cdots, s_k^i\}$ and its corresponding set of central points $p_c = \{c_1, c_2, \cdots, c_k\}$ can be obtained. Similarly, the set of LMPC $q = \{s_1^i, s_2^i, \cdots, s_K^i\}$ and the corresponding set of central points $q_c = \{c_1, c_2, \cdots, c_K\}$ can be obtained. In this study, three kinds of spheres with different radii were selected as multi-scale RSCS. After generating each sphere, it is also necessary to refer to some criteria for screening key points; the specific criteria can be found in [32]. Taking the rear floor beam of a car body as an example, the multi-scale RSCS of local measurement data on one side and the multi-scale RSCS of the digital model are shown in Figure 8. The different colours represent the point cloud covered by each divided sphere.

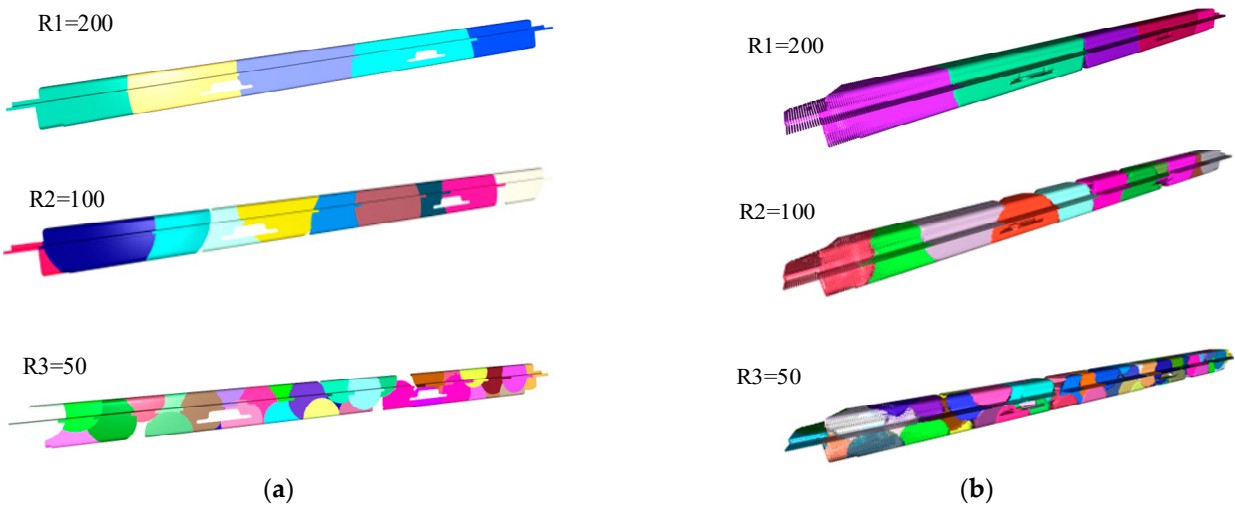

(**a**)             (**b**)

**Figure 8.** (**a**) multi-scale RSCS of measurement data; (**b**) multi-scale RSCS of CAD model.

### 3.2. Coarse Registration Based on Multivariate Descriptors (MD)

Based on spheres obtained from the above-mentioned method, each sphere's feature needs to be described. Feature descriptors are generally used for the initial alignment of the point cloud in the coarse registration stage. Common point cloud descriptors can be divided into global and local descriptors. Using a global descriptor requires that the size of the local measurement be much smaller than that of the global point cloud. However, some measurement data obtained for small workpieces are one-sided projections. When the

whole point cloud is divided according to the spherical scale of the local point cloud, the whole point cloud will be divided into a sphere, leading to registration failure. Therefore, point cloud local descriptors will be adopted in this study.

This study proposed a multivariate descriptor to improve registration accuracy. The combination of two classical descriptors: Fast Point Feature Histogram (FPFH) [33], and Shot descriptor, which is based on a local reference frame, is used in a subsequent experiment. The local descriptors used are not limited to the above two manual descriptors; it can also use some descriptors based on deep learning, such as Spinnet [34]. This study presented an optimised scheme to improve the description ability of descriptors. The descriptors do not belong to the research subject of this paper. Therefore, based on the hypothesis of the scheme, assuming that $n$ local feature descriptors are eventually used, each sphere will get n descriptor vectors. Coarse registration adopts sample consensus initial alignment (SAC-IA) [33], and the main contents of SAC-IA are as follows:

(1) The descriptor sets of multi-scale RSCS of LMPC and global point cloud are calculated, $\hat{p}_n = \left\{ d_1^i, d_2^i, \cdots, d_k^i \right\}, \hat{q}_n = \left\{ d_1^i, d_2^i, \cdots, d_K^i \right\}$, where $n$ represents different descriptors, $i$ is RSCS of different scales, $k$ and $K$ are the number of LMPC and whole CAD point cloud at a certain scale.

(2) $n$ descriptors are randomly selected from $\hat{p}_n$ and $\hat{q}_n$ to determine the $n$ corresponding relationship. The centre of RSCS is defined as the corresponding point. Supposing n descriptors in $\hat{p}_n$ determine the most similar corresponding $n$ points in $\hat{q}_n$, the weighted average points between the corresponding points of $n$ different descriptors are calculated as the final corresponding points. Then the final corresponding point position in $\hat{q}_n$ is defined by Equation (11), where $w_i$ is the normalised weight of the matching score in corresponding points from $n$ descriptor and $c_i$ is the position of the corresponding point from a different descriptor. Figure 9 is a sketch of a weighted average corresponding point with three descriptors, in which the blue is the local point cloud, and the grey is the whole DMPC. In the whole DMPC, different shapes correspond to the corresponding points calculated by different descriptors, and the circular points are the final weighted corresponding points.

(3) After determining the corresponding points, the corresponding points of the s group are randomly selected to obtain the corresponding point set $\hat{p}_c = \{c_1, c_2, \cdots, c_s\}$ and $\hat{q}_c = \{l_1, l_2, \cdots, l_s\}$, according to Equations (15) and (16), the transformation matrices $R$ and $T$ are calculated, where U and V are singular matrices; it is obtained by performing a singular value decomposition on Equation (14).

(4) Using the transformation matrix to carry out a rotation and translation transformation on corresponding points except for s corresponding points, if the distance of k-s corresponding point is less than a certain threshold value after transformation, then the point in the local point cloud is determined to be the inner point; otherwise, it is the outer point, the number of inner points is counted, and $s_{i+1}$ corresponding points are selected in the next round. Repeat the above process. The result is the corresponding points with the most significant number of inner points.

(5) Comparing the point cloud registration results at different scales and selecting the group with the smallest error as a result.

$$l = \sum_{i=0}^{n} w_i c_i \tag{11}$$

$$\overline{x} = \frac{1}{n} \sum_{i=0}^{n} x_i \tag{12}$$

$$h = \frac{1}{n} \sum_{i=0}^{n} (x_i - \overline{x})(y_i - \overline{y}) \tag{13}$$

$$R = +V^T \tag{14}$$

$$T = \bar{y} - R\bar{x} \tag{15}$$

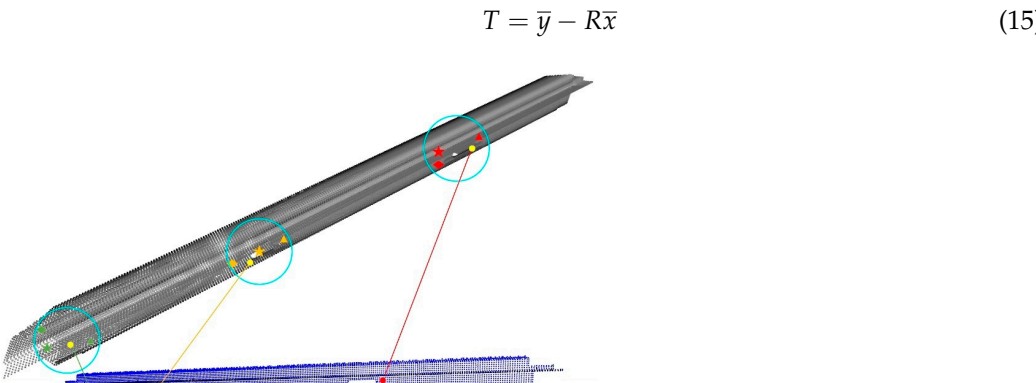

**Figure 9.** Weighted corresponding points of MD.

### 3.3. Iterative Closest Point (ICP) Based on Fine Registration

After the coarse registration of LMPC and DMPC is achieved via the above steps, we employ the iterative closest point (ICP) [35] algorithm to accurately match the scanned point cloud to the whole CAD model. In acceptable registration, all local point cloud is iterated, but it is easy to fall into local optimisation and time-consuming. Therefore, in this step, ICP iterates by finding the nearest point between the centre point of the RSCS sphere in the measurement point cloud and the target point cloud.

## 4. Local Point Cloud Segmentation

The surface topography analysis of all local measurement data can be conducted when the fine registration of LMPC and DMPC is completed. First, however, it is necessary to identify points and the functional surfaces associated with them. Let $P_c = \{P_1, P_2, \ldots, P_n\}$ be an LMPC, the determination of ownership between measuring points and the functional surface is through the Euclidean distance between the point and plane, then sorting the distance between the point and each surface and finding the closest surface is the corresponding functional surface. Before sorting, the point needs to be mapped to a 2D plane, and then the radial method is used to determine whether the point belongs to this plane. The distance from a point to the plane is expressed as Equation (16), where $n$ is the direction of the line, i.e., the normal direction of the plane, $p$ is a measurement point, and o is any point on the plane.

$$d = \frac{\|n(p - o)\|}{\|n\|} \tag{16}$$

Compared with the triangular surface in stereolithography (STL) files, the curved surface can be composed of multiple quadrilateral planes. Figure 10 illustrates the ownership judgement between a point and a curved surface, the point cloud is down-sampled to Figure 10b, and a plane is represented by four points. Let $P_s = \{P_1, P_2, \ldots, P_k\}$ be a set of curved surfaces, where $P_k \in P$ is the quadrilateral plane. Similar to the ownership judgement of plane, if a point belongs to any plane in the set $P$, then the point belongs to that surface (Figure 10c). Generally, the fewer planes on the surface, the higher the judgment error and the less time.

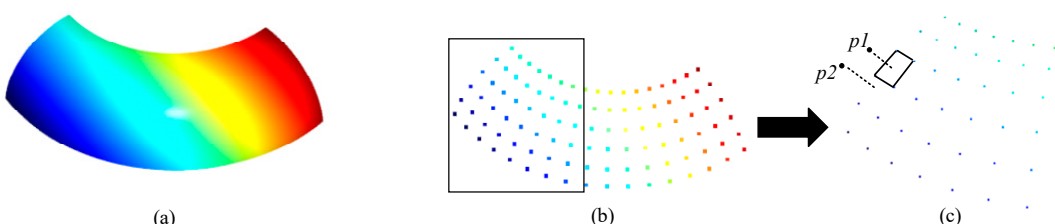

(a)          (b)          (c)

**Figure 10.** The judgement of whether a point belongs to a curved surface. (**a**) Curved surface; (**b**) Down-sample of point cloud; (**c**) The relation of points and quadrilateral plane.

## 5. Experimental Verification and Comparisons

### 5.1. Experimental Conditions

In this section, we perform experiments to validate the effectiveness of the proposed method. Ubuntu, PyCharm editor and Open3D library environment are used to analyse errors of workpiece surfaces. The flatness, cylindricity and surface roughness [36] of specific functional surfaces on two automobile body parts are examined. The experimental setup and workpieces are depicted in Figure 11 [37]. The schematic diagram of the 3D structured light measurement system is shown in Figure 12. The measured part is located on the reference surface X-Y, $O_c$ is the camera lens optical center, $O_p$ is the projector lens optical center, P is a point on the measured part, and its projection on the reference surface is P$'$, the length of P-P$'$ is h, points A and B are the intersection points between point P and the two optical centers and the reference plane, respectively, l and d are the distances from the optical center of the camera to the reference surface and the optical center of the projector. According to [31,38], the function of h and phase difference $\theta_A - \theta_B$ can be expressed as follows:

$$h = \frac{l(\theta_A - \theta_B)}{(\theta_A - \theta_B) + 2\pi d/\lambda_0} \tag{17}$$

where $\lambda_0$ is the optical grating pitch, $\theta_A$ and $\theta_B$ are the phase of points A and B, respectively.

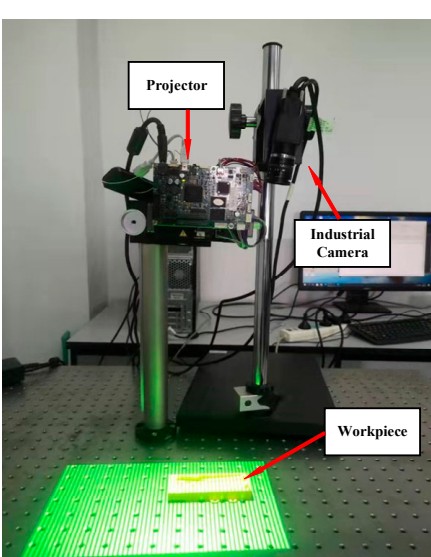

**Figure 11.** Structured light-based measurement system.

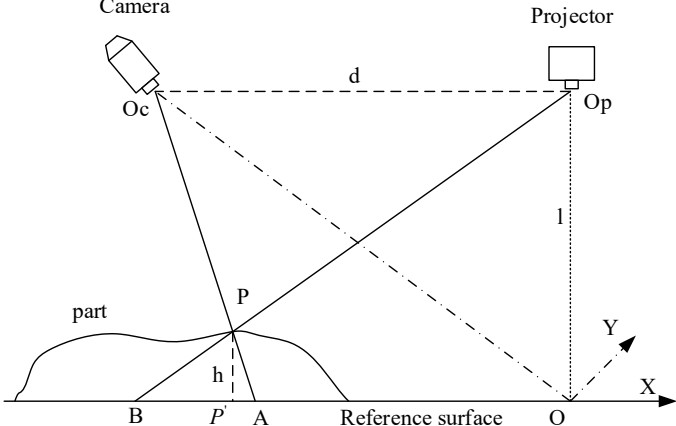

**Figure 12.** The optical path diagram of 3D structured light system.

STEP files of the two parts are discretised into a point cloud, as shown in Figure 13, where the LMPC is shown in the blue point cloud. Local measurement data are obtained by the structured light measurement system (accuracy is 0.05 mm). The local measurement data captured are all data on the upper surface of the workpiece.

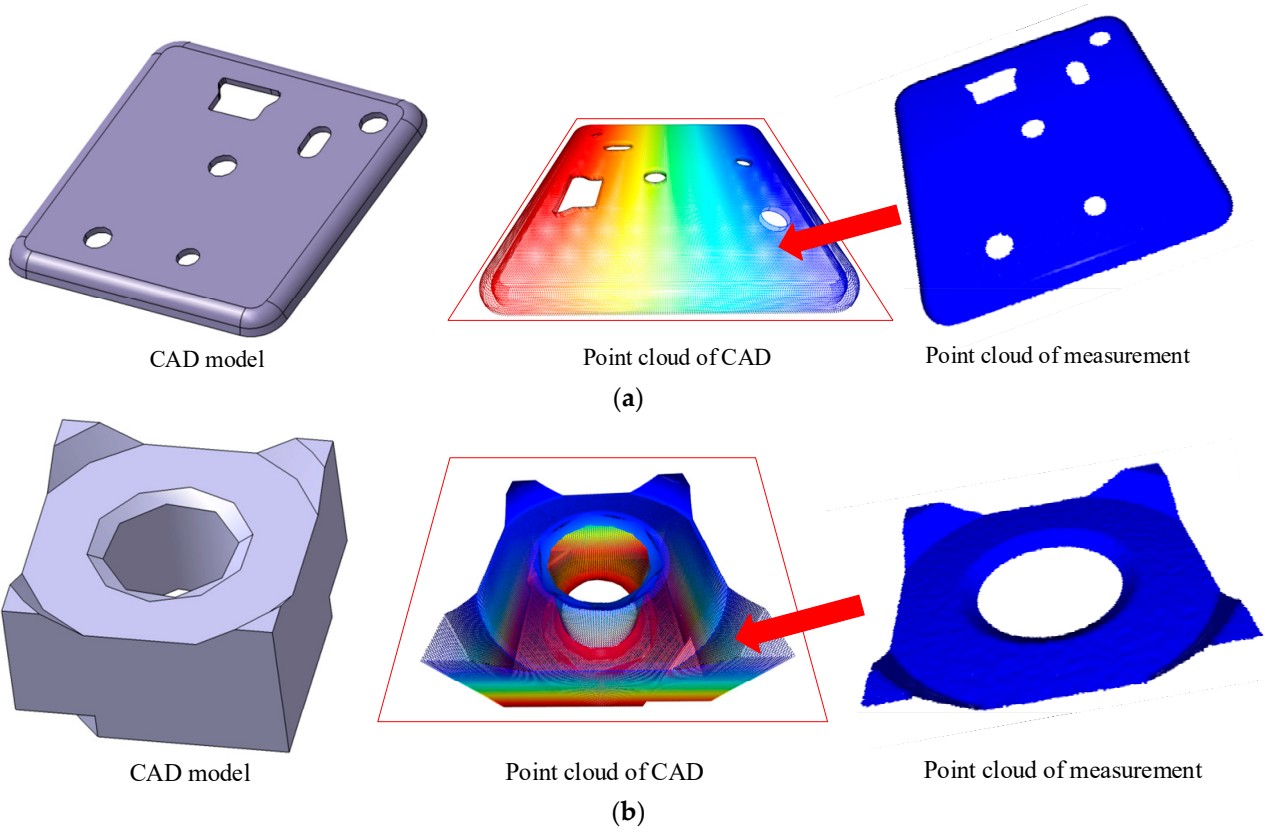

CAD model    Point cloud of CAD    Point cloud of measurement

(**a**)

CAD model    Point cloud of CAD    Point cloud of measurement

(**b**)

**Figure 13.** (**a**) Digital model point cloud and a measurement point cloud of part 1; (**b**) Digital model point cloud and a measurement point cloud of part 2.

### 5.2. The Analysis of Registration

In point cloud registration experiments, two point clouds are sampled with a voxel of size 1, and the registration error is calculated as

$$RMS = \frac{1}{n}\sqrt{\sum (p_i - f(q_i))^2} \tag{18}$$

where $p_i$ represents the target point cloud, $q_i$ represents the measurement point cloud, and $f$ represents the point cloud after registration. First, workpiece b is used to compare the registration accuracy of single-scale RSCS and multi-scale RSCS, and the results are shown in Table 1. Figure 14a–c shows the coarse registration results of spheres with scales of 20, 50 and 80, respectively. It can be seen that coarse registration results are relatively superior when the scale is 80, and using multiple scales can increase the accuracy of registration compared to a single scale. In general, the scale of the sphere in RSCS can be made according to the volume of the measured data. There is an empirical formula for setting a sphere's scale, expressed in Equation (19), where $V$ represents the volume of the minimal circumscribed sphere, and m is the number of spheres in RSCS.

**Table 1.** Influence of RSCS at different scales on parts registration error (mm).

| The Scale of RSCS | Registration Error |
|---|---|
| 20-scale RSCS+MD+ICP | 2.095 |
| 50-scale RSCS+MD+ICP | 4.233 |
| 80-scale RSCS+MD+ICP | 4.856 |

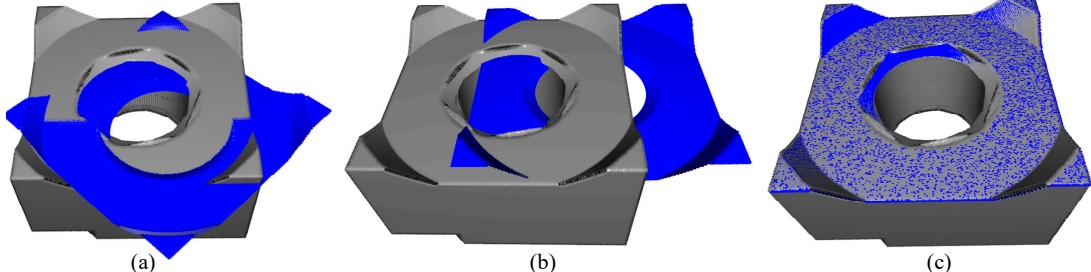

(a)  (b)  (c)

**Figure 14.** Registration results of RSCS at different scales. (**a**) Scale of 20; (**b**) Scale of 50; (**c**) Scale of 80.

The registration results of parts 1 and 2 are shown in Figures 15 and 16 and Table 2. The labels a and b represent the coarse and fine registration results. Table 2, "*part number_registration scheme*", shows the results of different registration schemes with different parts. Although only two descriptors are used in the multivariate descriptors in the experiment, the result is better than using a single descriptor. In the proposed coarse algorithm, the matching relation of the point cloud patch in the final RSCS is shown in Figure 16, where the black point is the weighted average corresponding point, the different colour of the point cloud patch is the sphere of the final RSCS, the line represents the corresponding relation. Although the proposed scheme can reduce the registration error and increase the probability of registration success, it can be seen from the broken line figure in Figure 17 that with the increase of sphere candidate scale and descriptor, especially with the rise of sphere candidate scale, the time consumption presents an exponential growth, so it needs a tradeoff in terms of actual use.

$$Radius = \sqrt[3]{\left(\frac{3}{4\pi} * \frac{0.7}{2m} * V\right)} \tag{19}$$

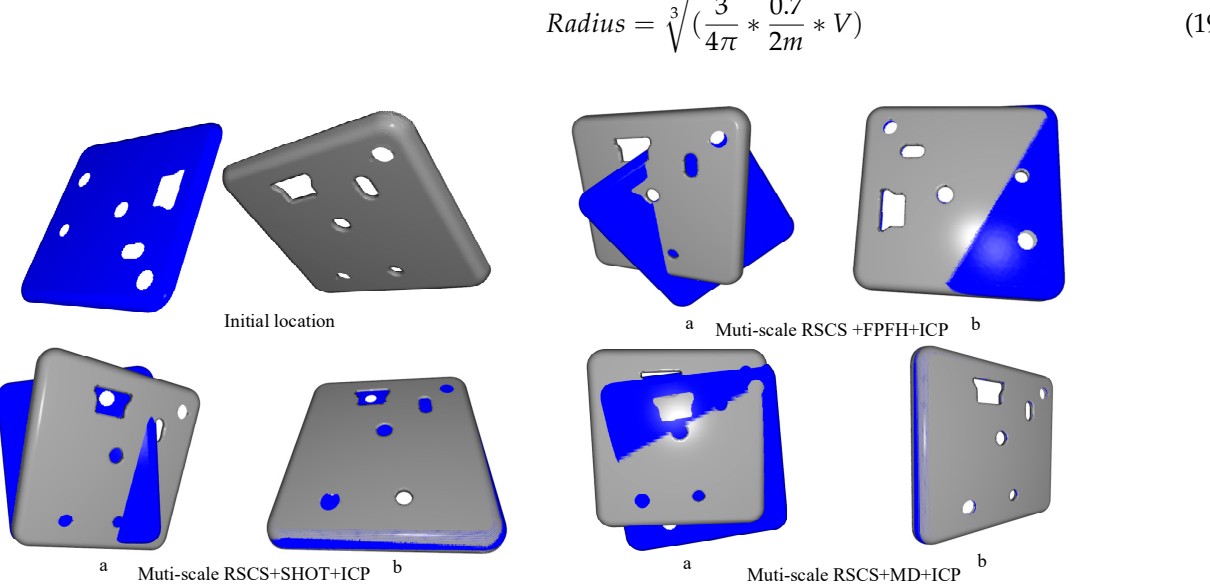

Initial location

a  Muti-scale RSCS +FPFH+ICP  b

a  Muti-scale RSCS+SHOT+ICP  b

a  Muti-scale RSCS+MD+ICP  b

**Figure 15.** Results of different registration schemes for part 1: (**a**) coarse registration; (**b**) fine registration.

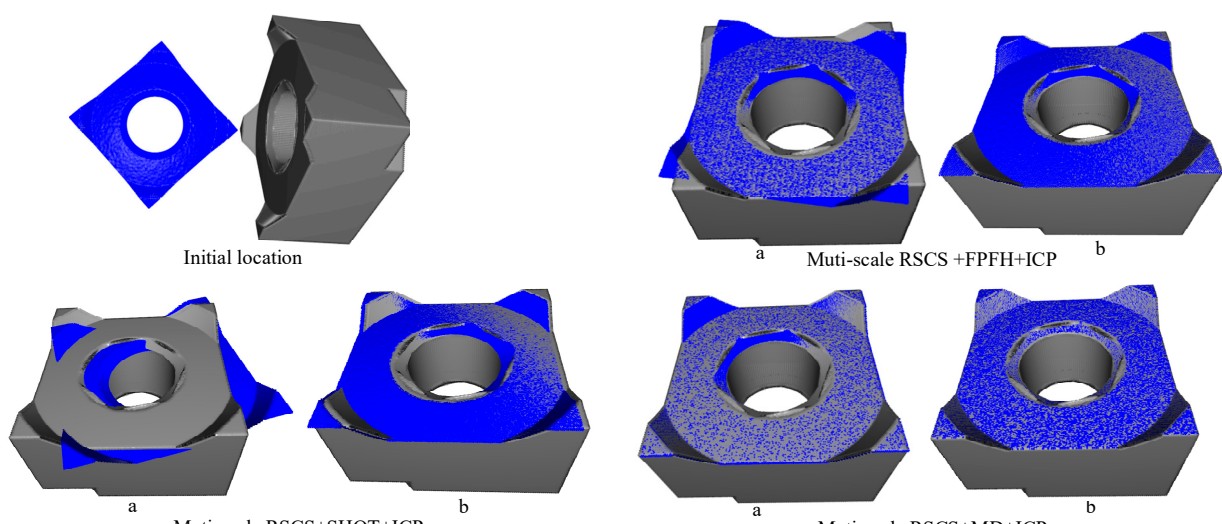

**Figure 16.** Results of different registration schemes for part 2: (**a**) coarse registration; (**b**) fine registration.

**Table 2.** Registration errors of different descriptors (mm).

| The Type of Descriptor | Registration Error |
|:---:|:---:|
| a_Muti-scale RSCS+SHOT+ICP | 6.499 |
| a_Muti-scale RSCS+FPFH+ICP | 6.211 |
| a_Muti-scale RSCS+MD+ICP | 4.279 |
| b_Muti-scale RSCS+SHOT+ICP | 4.556 |
| b_Muti-scale RSCS+FPFH+ICP | 6.988 |
| b_Muti-scale RSCS+MD+ICP | 2.095 |

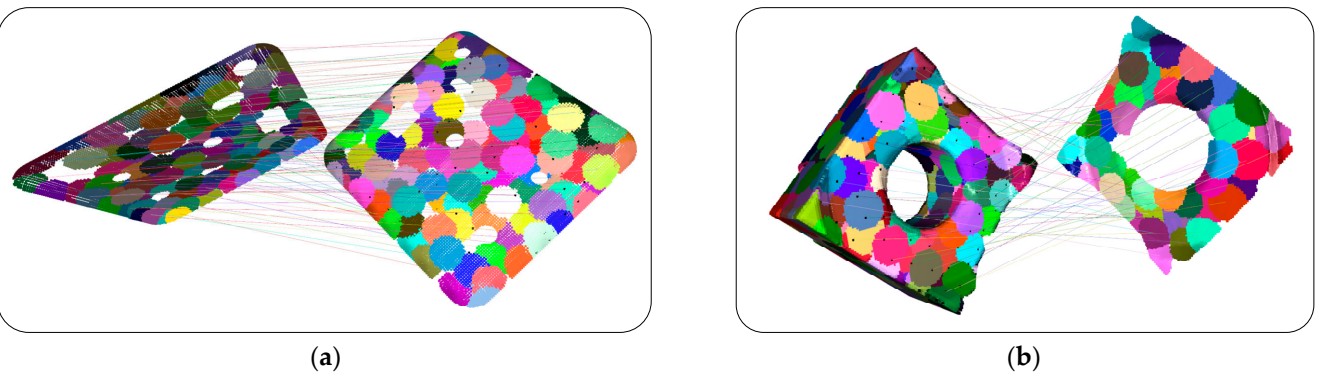

**Figure 17.** The matching relation of point cloud patch in Multi-scale RSCS+MD (**a**) part 1; (**b**) part 2.

*5.3. Point Cloud Segmentation and Error Analysis*

According to the template of each surface in the STEP file, each functional surface is segmented, as shown in Figure 18 where different colors represent different functional surfaces. Error analysis is conducted on specific functional surfaces of the measurement data on part 1 and part 2, respectively. The functional surfaces to be analysed were marked as a and b in Figure 18. The cylindricity, flatness and roughness of part 1 are analysed, and the flatness and roughness of part 2 are analysed. The results are shown in Table 3, where the roughness refers to ISO 25,178 [39] series and ISO 16,610 [36] series, the surface roughness was obtained by a two-dimensional Gaussian filter, and its sq parameters are calculated. The calculation method of flatness is as follows: First, the plane is fitted, and the linear equations shown in Equation (20) are solved, where $x_i$, $y_i$ and $z_i$ represent the three coordinates of the points in the surface point cloud, a, b and c represent the three

parameters in the general equation of the plane. After obtaining the plane equation, the distance between the points and the plane is calculated to obtain the distance set d, and the flatness is equal to $max(d) - min(d)$. The calculation method of cylindricity [40] is similar to flatness. A cylinder can be regarded as composed of radius R and axis l. The parameters to be fitted become the direction vector N ($n_x$, $n_y$, $n_z$) of the line, the origin point $o(o_x, o_y, o_z)$ of the line, and the radius R. Supposition a point pi ($x_i$, $y_i$, $z_i$) in the measurement point cloud, then establish the least squares model according to Equation (21) and then calculate the partial derivative of $st$ to each parameter to solve the least squares cylinder. The set of distances between each point and the axis is $d$, and cylindricity is $max(d) - min(d)$. According to the above error analysis method, the calculation results of each functional surface in Figure 18 are shown in Table 3. Through the STEP file, the ideal model of each functional surface can be obtained. The ideal model of S1 and S3 in Figure 18a is a cylinder, so its cylindricity can be calculated. S2 is a plane, and S1 and S2 in Figure 18b are also planes.

$$
\begin{bmatrix}
\sum x_i^2 & \sum x_i y_i & \sum x_i \\
\sum x_i y_i & \sum y_i^2 & \sum y_i \\
\sum x_i & \sum y_i & n
\end{bmatrix}
\begin{bmatrix}
a \\ b \\ c
\end{bmatrix}
=
\begin{bmatrix}
\sum x_i z_i \\
\sum y_i z_i \\
\sum z_i
\end{bmatrix}
\tag{20}
$$

$$
st = \left( r_i - \frac{1}{N} \sum_{i=0}^{N} r_i \right)^2
\tag{21}
$$

$$
r_i = |op_i \times n|
\tag{22}
$$

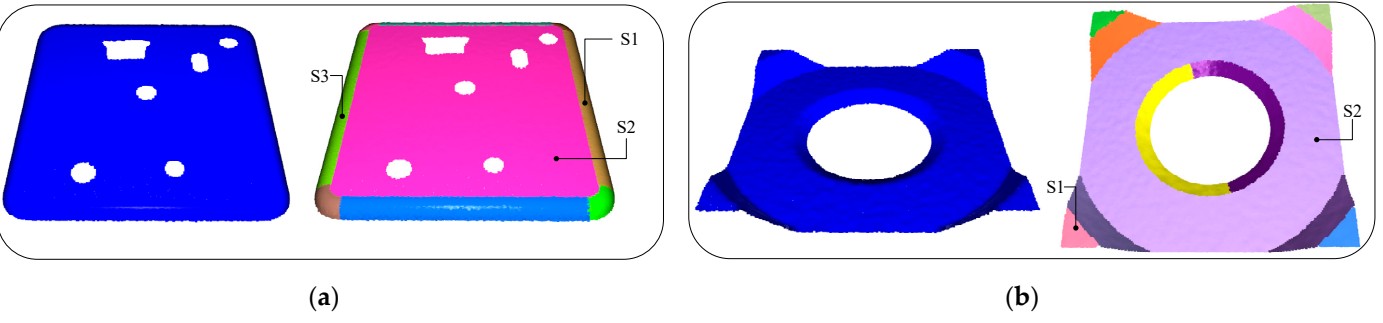

**Figure 18.** Part measurement point cloud segmentation results and the functional surface to be analysed. (**a**) Functional surfaces of part 1; (**b**) Functional surfaces of part 2.

**Table 3.** Error analysis for workpieces (mm).

| Features | Flatness | Cylindricity | $s_q$ Parameter in Roughness |
|---|---|---|---|
| a_S1 | - | 0.881 | 0.219 |
| a_S2 | 1.037 | - | 1.298 |
| a_S3 | - | 0.945 | 0.207 |
| b_S1 | 0.713 | - | 0.813 |
| b_S2 | 0.545 | - | 0.762 |

## 6. Discussion

Several surface topography evaluation methods are applied to evaluate the parts' errors; their results are shown in Tables 4–6 for comparison. In the comparison, the micrometer, CMM, and profilometer are selected.

**Table 4.** Results of comparison with existing methods for flatness (mm).

| Features | Micrometer | CMM | Profilometer | Proposed Method |
|---|---|---|---|---|
| a_S2 | 1.067 | 1.011 | 1.103 | 1.037 |
| b_S1 | 0.842 | 0.702 | 0.801 | 0.713 |
| b_S2 | 0.539 | 0.578 | 0.612 | 0.545 |

**Table 5.** Results of comparison with existing methods for cylindricity (mm).

| Features | Micrometer | CMM | Profilometer | Proposed Method |
|---|---|---|---|---|
| a_S1 | 0.892 | 0.856 | 1.020 | 0.881 |
| a_S3 | 1.035 | 0.921 | 1.142 | 0.945 |

**Table 6.** Results of comparison with existing methods for roughness (mm).

| Features | Micrometer | CMM | Profilometer | Proposed Method |
|---|---|---|---|---|
| a_S1 | 0.216 | 0.301 | 0.203 | 0.219 |
| a_S2 | 1.354 | 1.326 | 1.276 | 1.298 |
| a_S3 | 0.235 | 0.244 | 0.198 | 0.207 |
| b_S1 | 0.873 | 0.854 | 0.801 | 0.813 |
| b_S2 | 0.852 | 0.832 | 0.732 | 0.762 |

As illustrated in Tables 4–6, the results of the micrometer, CMM and profilometer are consistent with the results of the proposed method, and all of the evaluated values differ slightly. This demonstrates the feasibility and effectiveness of the proposed method in solving error evaluation with a 3D structured light system. Furthermore, the comparison has verified that the proposed method has high precision and can be employed in the automotive part measurement process.

### 7. Conclusions

This study presents a multi-component local descriptor method to optimize coarse registration precision. First, they obtain the measurement and digital model point cloud, further divided by the multi-scale RSCS. Then, multivariate descriptors were integrated to generate feature vectors, and the SAC-IA algorithm was applied to find the corresponding points, followed by the matching score to get the final position of the related points, and an ICP-based fine point cloud registration was carried out. Finally, the STEP file is applied to guide the measurement point cloud segmentation to complete the automatic error analysis of specific functional surfaces. In the experiment, this study selects the rear beam parts of an automobile for error analysis. One of the results is that the r-angle cylindricity is 0.881 mm, the roughness sq parameter is 0.219 mm, and the flatness of its plane is 1.037 mm. Compared with a micrometer, CMM and profilometer, the measurement results differ slightly; the results show that the proposed method can solve the challenging problem of automating error analysis based on optical measurement data.

The major contribution of this study is the development of a multivariate local descriptor registration method and segmentation method, which can be applied in a 3D structured light measurement system. The developed methods consider 3D surface topography measurement and improve registration accuracy compared to existing methods. Further, the methods are suitable for automatic error evaluation in the automobile industry when measurement for vehicle parts needs cost consideration.

The STEP file contains geometric information, which can not only serve as the guidance of point cloud segmentation but also complete some feature recognition and the mid-surface generation algorithm of CAE. However, although the proposed descriptors can improve registration precision, there are still registration failures for minimal shapes. Therefore, deep learning-based descriptors will be considered in the registration positioning stage to improve the registration robustness further in future work.

**Author Contributions:** Conceptualization, C.K., Y.X. and Z.L.; Software, J.L.; Validation, Z.L., J.L. and I.M.; Investigation, C.K. and C.Z.; Resources, I.M. and Y.X.; Writing—original draft, C.K. and Z.L.; Writing—review & editing, Y.X. and T.L.; Visualization, C.K. and D.T.; Supervision, Y.X. and X.J.; Funding acquisition, Y.X. All authors have read and agreed to the published version of the manuscript.

**Funding:** This research was funded by the National Natural Science Foundation of China (NSFC) (61203172); the Sichuan Science and Technology Programs (2023NSFSC0361, 2022002).

**Informed Consent Statement:** Informed consent was obtained from all subjects involved in the study.

**Data Availability Statement:** All data generated or analyzed to support the findings of the present study are included in this article. The raw data can be obtained from authors, upon reasonable request.

**Conflicts of Interest:** The authors declare no conflict of interest.

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
