# Peer review of "A Multivariate Local Descriptor Registration Method for Surface Topography Evaluation"

_applsci, doi:10.3390/app13053311_

Round 1

Reviewer 1 Report

The issues I have mentioned below should be clarified. It is acceptable after these corrections are made.

1) There is no detailed explanation about the “3D structured light measurement system” measurement logic you have used. Which measurement technique does it work with?

2) What are the calibration values of “3D structured light measurement system”. How much these calibration values affect the accuracy of your measurement.

3) You mentioned the previous studies very briefly. You need to increase this part a little more. You should briefly mention the results of these studies. (Studies must be current studies)

4) Which digital camera “3D structured light measurement system” did you use. What is the resolution of this camera? What would your results be if you used a higher resolution camera while running these tests?

5) The discussion section can be added to the article. In this section, you can compare your own results with similar studies.

6) Are the values you get in the results section (the rangle cylindricity is 0.881mm, the roughness sq parameter is 0.219mm, and the flatness of its plane is 1.037mm.) really sensitive. Did you reference any value?

7) the literature should be increased.

Reviewer 2 Report

Dear Authors,

the article is scientifically interesting and has publication potential. I suggest making the following adjustments to improve the paper:

The introduction section - first paragraph - describes the optical methods of 3D surface topography measurement, such as CT and CMM. The basic method, which is 3D optical microscopy, is not given. Please add this technique with your comment.

Figure 1 - enter the units on the line with numerical values

In the introduction and conclusions, clearly state the novelty of this paper. What has already been done in this topic, and what new authors propose in their research.

The STEP file contain rich geometric data, and all surface shapes are conformable to the initial design intentions of CAD designers” - What do the authors mean by “rich geometric” data?

Equation (4) looks unfinished

What do the colors in Figure 5 mean? Please describe in detail.

Subscripts are not written everywhere in the text. Sometimes these are large numbers, sometimes subscripts, so there is no uniformity in the notation.

In table 1, it is better to arrange the data in ascending or descending order. There is no order at the moment.

Please describe the surface parameter Sq that appeared in the text without much explanation

Please write more about multiscale methods in the introduction. Describe several multiscale analysis techniques. Information about this can be found in publications:

Length- and Area-Scale Analyses - https://doi.org/10.3390/cryst11111371

Review of multiscale analyzes - theory and applications - https://doi.org/10.1016/j.cirp.2018.06.001

Macro- meso- micro-scale analyzes - https://doi.org/10.1016/j.ijmecsci.2021.106808
